# Neural Network Approximation based on Hausdorff distance of Tropical Zonotopes

**Panagiotis Misiakos**[1, †]**, Georgios Smyrnis**[2]**, Georgios Retsinas**[3]**, Petros Maragos**[3]

[1]ETH Zurich, [2]University of Texas at Austin, [3]National Technical University of Athens
`panagiotis.misiakos@inf.ethz.ch, gsmyrnis@utexas.edu,`
`gretsinas@central.ntua.gr, maragos@cs.ntua.gr`

## Abstract

In this work we theoretically contribute to neural network approximation by providing a novel tropical geometrical viewpoint to structured neural network compression. In particular, we show that the approximation error between two neural networks with ReLU activations and one hidden layer depends on the Hausdorff distance of the tropical zonotopes of the networks. This theorem comes as a first step towards a purely geometrical interpretation of neural network approximation. Based on this theoretical contribution, we propose geometrical methods that employ the K-means algorithm to compress the fully connected parts of ReLU activated deep neural networks. We analyze the error bounds of our algorithms theoretically based on our approximation theorem and evaluate them empirically on neural network compression. Our experiments follow a proof-of-concept strategy and indicate that our geometrical tools achieve improved performance over relevant tropical geometry techniques and can be competitive against non-tropical methods.

## 1 Introduction

Tropical geometry (Maclagan & Sturmfels, 2015) is a mathematical field based on algebraic geometry and strongly linked to polyhedral and combinatorial geometry. It is built upon the *tropical semiring* which originally refers to the min-plus semiring $(\mathbb{R}_{\min}, \wedge, +)$, but may also refer to the max-plus semiring (Cuninghame-Green, 2012; Butkovič, 2010). In our work, we follow the convention of the max-plus semiring $(\mathbb{R}_{\max}, \vee, +)$ which replaces the classical operations of addition and multiplication by max and sum respectively. These operations turn polynomials into piecewise linear functions making them directly applicable in neural networks.

Tropical mathematics cover a wide range of applications including dynamical systems on weighted lattices (Maragos, 2017), finite state transducers (Theodosis & Maragos, 2018; 2019) and convex regression (Maragos & Theodosis, 2020; Tsilivis et al., 2021). Recently, there has been remarkable theoretical impact of tropical geometry in the study of neural networks and machine learning (Maragos et al., 2021). Zhang et al. (2018) prove the equivalence of ReLU activated neural networks with tropical rational mappings. Furthermore, they use zonotopes to compute a bound on the number of the network's linear regions, which has already been known in (Montúfar et al., 2014). In a similar context, Charisopoulos & Maragos (2018) compute an upper bound to the number of linear regions of convolutional and maxout layers and propose a randomized algorithm for linear region counting. Other works employ tropical geometry to examine the training and further properties of morphological perceptron (Charisopoulos & Maragos, 2017) and morphological neural networks (Dimitriadis & Maragos, 2021).

Pruning or, generally, compressing neural networks gained interest in recent years due to the surprising capability of reducing the size of a neural network without compromising performance (Blalock et al., 2020). As tropical geometry explains the mathematical structure of neural networks, pruning may also be viewed under the perspective of tropical geometry. Indeed, Alfarra et al. (2020) propose an unstructured compression algorithm based on sparsifying the zonotope matrices of the network. Also,

---

[†]Conducted research as a student in National Technical University of Athens.

Smyrnis et al. (2020) construct a novel tropical division algorithm that applies to neural network minimization. A generalization of this applies to multiclass networks (Smyrnis & Maragos, 2020).

**Contributions**    In our work, we contribute to structured neural network approximation from the mathematical viewpoint of tropical geometry:

- We establish a novel bound on the approximation error between two neural networks with ReLU activations and one hidden layer. To prove this we bound the difference of the networks' tropical polynomials via the Hausdorff distance of their respective zonotopes.
- We construct two geometrical neural network compression methods that are based on zonotope reduction and employ K-means algorithm for clustering. Our algorithms apply on the fully connected layers of ReLU activated neural networks.
- Our algorithms are analyzed both theoretically and experimentally. The theoretical evaluation is based on the theoretical bound of neural network approximation error. On the experimental part, we examine the performance of our algorithms on retaining the accuracy of convolutional neural networks when applying compression on their fully connected layers.

## 2    BACKGROUND ON TROPICAL GEOMETRY

We study tropical geometry from the viewpoint of the *max-plus semiring* $(\mathbb{R}_{\max}, \vee, +)$ which is defined as the set $\mathbb{R}_{\max} = \mathbb{R} \cup \{-\infty\}$ equipped with two operations $(\vee, +)$. Operation $\vee$ stands for max and $+$ stands for sum. In max-plus algebra we define polynomials in the following way.

**Tropical polynomials**    A *tropical polynomial* $f$ in $d$ variables $\mathbf{x} = (x_1, x_2, ..., x_d)^T$ is defined as the function

$$f(\mathbf{x}) = \max_{i \in [n]}\{\mathbf{a}_i^T \mathbf{x} + b_i\} \tag{1}$$

where $[n] = \{1, ..., n\}$, $\mathbf{a}_i$ are vectors in $\mathbb{R}^d$ and $b_i$ is the corresponding monomial coefficient in $\mathbb{R}_{\max} = \mathbb{R} \cup \{-\infty\}$. The set of such polynomials constitutes the semiring $\mathbb{R}_{\max}[\mathbf{x}]$ of tropical polynomials. Note that each term $\mathbf{a}_i^T \mathbf{x} + b_i$ corresponds to a hyperplane in $\mathbb{R}^d$. We thus call the vectors $\{\mathbf{a}_i\}_{i \in [n]}$ the *slopes* of the tropical polynomial, and $\{b_i\}_{i \in [n]}$ the respective *biases*. We allow slopes to be vectors with real coefficients rather than integer ones, as it is normally the case for polynomials in regular algebra. These polynomials are also referred to as *signomials* (Duffin & Peterson, 1973) in the literature.

**Polytopes**    Polytopes have been studied extensively (Ziegler, 2012; Grünbaum, 2013) and occur as a geometric tool for fields such as linear programming and optimization. They also have an important role in the analysis of neural networks. For instance, Zhang et al. (2018); Charisopoulos & Maragos (2018) show that linear regions of neural networks correspond to vertices of polytopes. Thus, the counting of linear regions reduces to a combinatorial geometry problem. In what follows, we explore this connection of tropical geometry with polytopes.

Consider the tropical polynomial defined in (1). The *Newton polytope* associated to $f(\mathbf{x})$ is defined as the convex hull of the slopes of the polynomial

$$\text{Newt}\,(f) := \text{conv}\{\mathbf{a}_i : i \in [n]\}$$

Furthermore, the *extended Newton polytope* of $f(\mathbf{x})$ is defined as the convex hull of the slopes of the polynomial extended in the last dimension by the corresponding bias coefficient.

$$\text{ENewt}\,(f) := \text{conv}\{(\mathbf{a}_i^T, b_i) : i \in [n]\}$$

The following proposition computes the extended Newton polytope that occurs when a tropical operation is applied between two tropical polynomials. It will allow us to compute the polytope representation corresponding to a neural network's hidden layer.

**Proposition 1.** *(Zhang et al., 2018; Charisopoulos & Maragos, 2018) Let $f, g \in \mathbb{R}_{max}[\mathbf{x}]$ be two tropical polynomials . Then for the extended Newton polytopes it is true that*

$$ENewt\,(f \vee g) = conv\{ENewt\,(f) \cup ENewt\,(g)\}$$
$$ENewt\,(f + g) = ENewt\,(f) \oplus ENewt\,(g)$$

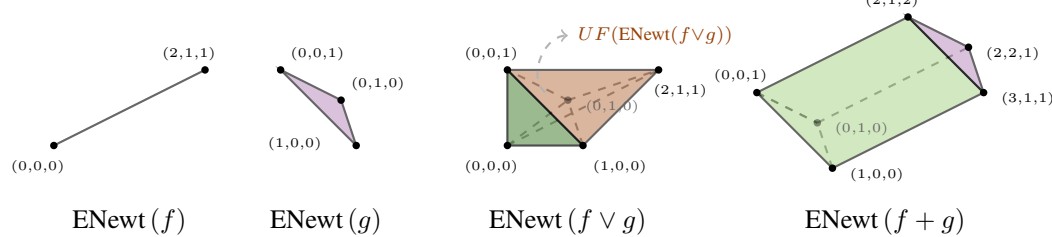

Figure 1: Illustration of tropical operations between polynomials. The polytope of the max ($\vee$) of $f$ and $g$ corresponds to the convex hull of the union of points of the two polytopes and the polytope of sum ($+$) corresponds to their Minkowski sum.

Here $\oplus$ denotes Minkowski addition. In particular, for two sets $A$, $B \subseteq \mathbb{R}^d$ it is defined as

$$A \oplus B := \{\mathbf{a} + \mathbf{b} \mid \mathbf{a} \in A, \mathbf{b} \in B\}$$

**Corollary 1.** *This result can be extended to any finite set of polynomials using induction.*

**Example 1.** *Let $f, g$ be two tropical polynomials in 2 variables, such that*

$$f(x, y) = \max(2x + y + 1, 0), \quad g(x, y) = \max(x, y, 1)$$

*The tropical operations applied to these polynomials give*

$$f \vee g = \max(2x + y + 1, 0, x, y, 1)$$
$$f + g = \max(3x + y + 1, x, 2x + 2y + 1, y, 2x + y + 2, 1)$$

*Fig. 1 illustrates the extended Newton polytopes of the original and the computed polynomials.*

The extended Newton polytope provides a geometrical representation of a tropical polynomial. In addition, it may be used to compute the values that the polynomial attains, as Proposition 2 indicates.

**Proposition 2.** *(Charisopoulos & Maragos, 2018) Let $f \in \mathbb{R}_{max}[\mathbf{x}]$ be a tropical polynomial in $d$ variables. Let $UF(ENewt(f))$ be the points in the upper envelope of $ENewt(f)$, where upward direction is taken with respect to the last dimension of $\mathbb{R}^{d+1}$. Then for each $i \in [n]$ there exists a linear region of $f$ on which $f(\mathbf{x}) = \mathbf{a}_i^T \mathbf{x} + b_i$ if and only if $(\mathbf{a}_i^T, b_i)$ is a vertex of $UF(ENewt(f))$.*

**Example 2.** *Using the polynomials from Example 1 we compute a reduced representation for $f \vee g$.*

$$f \vee g = \max(2x + y + 1, 0, x, y, 1) = \max(2x + y + 1, x, y, 1)$$

*Indeed, the significant terms correspond to the vertices of $UF(ENewt(f \vee g))$ shown in Fig. 1.*

## 2.1 Tropical Geometry of Neural Networks

Tropical geometry has the capability of expressing the mathematical structure of ReLU activated neural networks. We review some of the basic properties of neural networks and introduce notation that will be used in our analysis. For this purpose, we consider the ReLU activated neural network of Fig. 2 with one hidden layer.

**Network tropical equations** The network of Fig. 2 consists of an *input layer* $\mathbf{x} = (x_1, ..., x_d)$, a *hidden layer* $f = (f_1, ..., f_n)$ with ReLU activations, an output layer $v = (v_1, ..., v_m)$ and two linear layers defined by the matrices $A, C$ respectively. As illustrated in Fig. 2 we have $A_{i,:} = (\mathbf{a}_i^T, b_i)$ for the first linear layer and $C_{j,:} = (c_{j1}, c_{j2}, ..., c_{jn})$ for the second linear layer, as we ignore its biases. Furthermore, the output of the $i-$th component of the hidden layer $f$ is computed as

$$f_i(\mathbf{x}) = \max\left(\sum_{k=1}^d a_{ik} x_k + b_i, 0\right) = \max(\mathbf{a}_i^T \mathbf{x} + b_i, 0) \tag{2}$$

We deduce that each $f_i$ is a tropical polynomial with two terms. It therefore follows that $ENewt(f_i)$ is a linear segment in $\mathbb{R}^{d+1}$. The components of the output layer may be computed as

$$v_j(\mathbf{x}) = \sum_{i=1}^n c_{ji} f_i(\mathbf{x}) = \sum_{c_{ji}>0} |c_{ji}| f_i(\mathbf{x}) - \sum_{c_{ji}<0} |c_{ji}| f_i(\mathbf{x}) = p_j(\mathbf{x}) - q_j(\mathbf{x}) \tag{3}$$

**Tropical rational functions** In Eq. (3), functions $p_j, q_j$ are both linear combinations of $\{f_i\}$ with positive coefficients, which implies that they are tropical polynomials. We conclude that every output node $v_i$ can be written as a difference of two tropical polynomials, which is defined as a *tropical rational function*. This indicates that the output layer of the neural network of Fig. 2 is equivalent to a *tropical rational mapping*. In fact, this result holds for deeper networks, in general, as demonstrated by the following theorem.

**Theorem 1.** (Zhang et al., 2018) *A ReLU activated deep neural network $F : \mathbb{R}^d \to \mathbb{R}^m$ is equivalent to a tropical rational mapping.*

It is not known whether a tropical rational function $r(\mathbf{x})$ admits an efficient geometric representation that determines its values $\{r(\mathbf{x})\}$ for $\mathbf{x} \in \mathbb{R}^d$, as it holds for tropical polynomials with their polytopes in Proposition 2. For this reason, we choose to work separately on the polytopes of the tropical polynomials $p_j, q_j$.

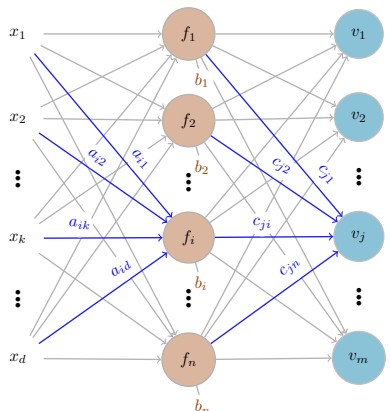

Figure 2: Neural network with one hidden ReLU layer. The first linear layer has weights $\{\mathbf{a}_i^T\}$ with bias $\{b_i\}$ corresponding to $i$−th node $\forall i \in [n]$ and the second has weights $\{c_{ji}\}, \forall j \in [m], i \in [n]$.

**Zonotopes** Zonotopes are defined as the Minkowski sum of a finite set of line segments. They are a special case of polytopes that occur as a building block for our network. These geometrical structures provide a representation of the polynomials $p_j, q_j$ in (3) that further allows us to build our compression algorithms. We use the notation $P_j, Q_j$ for the extended Newton polytopes of tropical polynomials $p_j, q_j$, respectively. Notice from (3) that for each component $v_j$ of the output $p_j, q_j$ are written as linear combinations of tropical polynomials that correspond to linear segments. Thus $P_j$ and $Q_j$ are zonotopes. We call $P_j$ the *positive* zonotope, corresponding to the positive polynomial $p_j$ and $Q_j$ the *negative* one.

**Zonotope Generators** Each neuron of the hidden layer represents geometrically a line segment contributing to the positive or negative zonotope. We thus call these line segments *generators* of the zonotope. The generators further receive the characterization *positive* or *negative* depending on the zonotope they contribute to. It is intuitive to expect that a zonotope gets more complex as its number of generators increases. In fact, each vertex of the zonotope can be computed as the sum of vertices of the generators, where we choose a vertex from each generating line segment, either $\mathbf{0}$ or $c_{ji} \left(\mathbf{a}_i^T, b_i\right)$. We summarize the above with the following extension of (Charisopoulos & Maragos, 2018).

**Proposition 3.** $P_j, Q_j$ *are zonotopes in $\mathbb{R}^{d+1}$. For each vertex $\mathbf{v}$ of $P_j$ there exists a subset of indices $I_+$ of $\{1, 2, ..., n\}$ with $c_{ji} > 0, \forall i \in I_+$ such that $\mathbf{v} = \sum_{i \in I_+} c_{ji} \left(\mathbf{a}_i^T, b_i\right)$. Similarly, a vertex $\mathbf{u}$ of $Q_j$ can be written as $\mathbf{u} = \sum_{i \in I_-} c_{ji} \left(\mathbf{a}_i^T, b_i\right)$ where $I_-$ corresponds to $c_{ji} < 0, \forall i \in I_-$.*

## 3 APPROXIMATION OF TROPICAL POLYNOMIALS

In this section we present our central theorem that bounds the error between the original and approximate neural network, when both have the architecture of Fig. 2. To achieve this we need to derive a bound for the error of approximating a simpler functional structure, namely the tropical polynomials that represent the neural network. The motivation behind the geometrical bounding of the error of the polynomials is Proposition 2. It indicates that a polynomial's values are determined at each point of the input space by the vertices of the upper envelope of its extended Newton polytope. Therefore, it is expected that two tropical polynomials with approximately equal extended Newton polytopes should attain similar values. In fact, this serves as the intuition for our theorem. The metric we use to define the distance between extended Newton polytopes is the *Hausdorff distance*.

**Hausdorff distance** The distance of a point $\mathbf{u} \in \mathbb{R}^d$ from the finite set $\mathcal{V} \subset \mathbb{R}^d$ is denoted by either $\text{dist}(\mathbf{u}, \mathcal{V})$ or $\text{dist}(\mathcal{V}, \mathbf{u})$ and computed as $\min_{\mathbf{v} \in \mathcal{V}} \|\mathbf{u} - \mathbf{v}\|$ which is the Euclidean distance of $\mathbf{u}$

from its closest point $\mathbf{v} \in \mathcal{V}$. The *Hausdorff distance* $\mathcal{H}(\mathcal{V}, \mathcal{U})$ of two finite point sets $\mathcal{V}, \mathcal{U} \subset \mathbb{R}^d$ is defined as

$$\mathcal{H}(\mathcal{V}, \mathcal{U}) = \max \left\{ \max_{\mathbf{v} \in \mathcal{V}} \mathrm{dist}\,(\mathbf{v}, \mathcal{U}), \max_{\mathbf{u} \in \mathcal{U}} \mathrm{dist}\,(\mathcal{V}, \mathbf{u}) \right\}$$

Let $P, \tilde{P}$ be two polytopes with their vertex sets denoted by $\mathcal{V}_P, \mathcal{V}_{\tilde{P}}$ respectively. We define the Hausdorff distance $\mathcal{H}\left(P, \tilde{P}\right)$ of the two polytopes as the Hausdorff distance of their respective vertex sets $\mathcal{V}_P, \mathcal{V}_{\tilde{P}}$. Namely,

$$\mathcal{H}\left(P, \tilde{P}\right) := \mathcal{H}(\mathcal{V}_P, \mathcal{V}_{\tilde{P}}) \tag{4}$$

Clearly, the Hausdorff distance is a metric of how close two polytopes are to each other. Indeed, it becomes zero when the two polytopes coincide. According to this metric, the following bound on the error of tropical polynomial approximation is derived.

**Proposition 4.** *Let $p, \tilde{p} \in \mathbb{R}_{max}[\mathbf{x}]$ be two tropical polynomials and let $P = ENewt\,(p)\,, \tilde{P} = ENewt\,(\tilde{p})$. Then,*

$$\max_{x \in \mathcal{B}} |p(\mathbf{x}) - \tilde{p}(\mathbf{x})| \leq \rho \cdot \mathcal{H}\left(P, \tilde{P}\right)$$

*where $\mathcal{B} = \{\mathbf{x} \in \mathbb{R}^d : \|\mathbf{x}\| \leq r\}$ is the hypersphere of radius $r$, and $\rho = \sqrt{r^2 + 1}$.*

The above proposition enables us to handle the more general case of neural networks with one hidden layer, that are equivalent with tropical rational mappings. By repeatedly applying Proposition 4 to each tropical polynomial corresponding to the networks, we get the following bound.

**Theorem 1.** *Let $v, \tilde{v} \in \mathbb{R}_{max}[\mathbf{x}]$ be two neural networks with architecture as in Fig. 2. With $\tilde{P}_j, \tilde{Q}_j$ we denote the positive and negative zonotopes of $\tilde{v}$. The following bound holds.*

$$\max_{x \in \mathcal{B}} \|v(\mathbf{x}) - \tilde{v}(\mathbf{x})\|_1 \leq \rho \cdot \left( \sum_{j=1}^{m} \mathcal{H}\left(P_j, \tilde{P}_j\right) + \mathcal{H}\left(Q_j, \tilde{Q}_j\right) \right)$$

**Remark 1.** The reason we choose to compute the error of the approximation on a bounded hypersphere $\mathcal{B}$ is twofold. Firstly, the unbounded error of linear terms always diverges to infinity and, secondly, in practice the working subspace of our dataset is usually bounded, e.g. images.

## 4 NEURAL NETWORK COMPRESSION ALGORITHMS

**Compression problem formulation** The tropical geometry theorem on the approximation of neural networks enables us to derive compression algorithms for ReLU activated neural networks. Suppose that we want to compress the neural network of Fig. 2 by reducing the number of neurons in the hidden layer, from $n$ to $K$. Let us assume that the output of the compressed network is the tropical rational map $\tilde{v} = (\tilde{v}_1, ..., \tilde{v}_m)$. Its $j$−th component may be written as $\tilde{v}_j(\mathbf{x}) = \tilde{p}_j(\mathbf{x}) - \tilde{q}_j(\mathbf{x})$ where using Proposition 3 the zonotopes of $\tilde{p}_j, \tilde{q}_j$ are generated by $\tilde{c}_{ji}(\tilde{\mathbf{a}}_i^T, \tilde{b}_i), \forall i$. The generators need to be chosen in such a way that $\tilde{v}_j(\mathbf{x}) \approx v_j(\mathbf{x})$ for all $\mathbf{x} \in \mathcal{B}$. Due to Theorem 1 it suffices to find generators such that the resulting zonotopes have $\mathcal{H}\left(P_j, \tilde{P}_j\right), \mathcal{H}\left(Q_j, \tilde{Q}_j\right)$ as small as possible $\forall j$. We thus formulated neural network compression as a geometrical zonotope approximation problem.

**Our approaches** Approximating a zonotope with fewer generators is a problem known as zonotope order reduction (Kopetzki et al., 2017). In our case we approach this problem by manipulating the zonotope generators $c_{ji}\left(\mathbf{a}_i^T, b_i\right), \forall i, j$ [1]. Each of the algorithms presented will create a subset of altered generators that approximate the original zonotopes. Ideally, we require the approximation to hold simultaneously for all positive and negative zonotopes of each output component $v_j$. However, this is not always possible, as in the case of multiclass neural networks, and it necessarily leads to heuristic manipulation. Our first attempt to tackle this problem applies the K-means algorithm to the

---

[1] Dealing with the full generated zonotope would lead to exponential computational overhead.

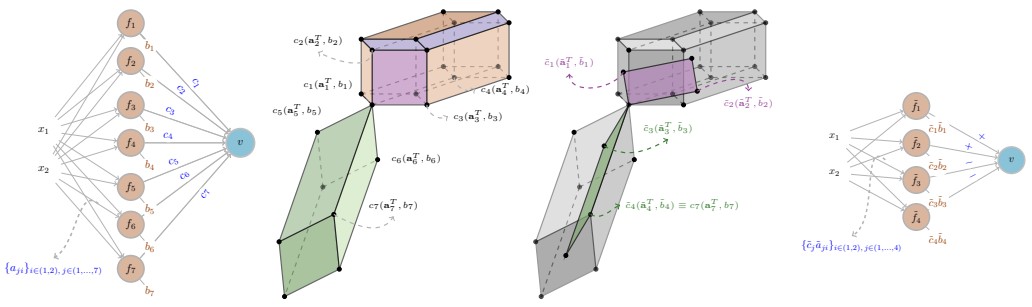

(a) Original network.   (b) Original zonotopes.   (c) Resulting zonotopes.   (d) Compressed network.

Figure 3: Illustration of Zonotope K-means execution. The original zonotope $P$ is generated by $c_i\left(\mathbf{a}_i^T,\, b_i\right)$ for $i = 1, ..., 4$ and the negative zonotope $Q$ generated by the remaining ones $i = 5, 6, 7$. The approximation $\tilde{P}$ of $P$ is colored in purple and generated by $\tilde{c}_i\left(\tilde{\mathbf{a}}_i^T,\, \tilde{b}_i\right)$, $i = 1, 2$ where the first generator is the K-means center representing the generators $1, 2$ of $P$ and the second is the representative center of $3, 4$. Similarly, the approximation $\tilde{Q}$ of $Q$ is colored in green and defined by the generators $\tilde{c}_i\left(\tilde{\mathbf{a}}_i^T,\, \tilde{b}_i\right)$, $i = 3, 4$ that stand as representative centers for $\{5, 6\}$ and 7 respectively.

positive and negative generators, separately. This method is restricted on applying to single output neural networks. Our second approach further develops this technique to multiclass neural networks. Specifically, it utilizes K-means on the vectors associated with the neural paths passing from a node in the hidden layer, as we define later. The algorithms we present refer to the neural network of Fig. 2 with one hidden layer, but we may repeatedly apply them to compress deeper networks.

### 4.1 ZONOTOPE APPROXIMATION

**Zonotope K-means**   The first compression approach uses K-means to compress each zonotope of the network, and covers only the case of a single output neural network, e.g as in Fig. 2 but with $m = 1$. The algorithm reduces the hidden layer size from $n$ to $K$ neurons. We use the notation $c_i$, $i = 1, ..., n$ for weights of the second linear layer, connecting the hidden layer with the output node. Algorithm 1 is presented below and a demonstration of its execution can be found in Fig. 3.

---

**Algorithm 1:** Zonotope K-means Compression

1. Split generators into positive $\left\{c_i\left(\mathbf{a}_i^T,\, b_i\right) : c_i > 0\right\}$ and negative $\left\{c_i\left(\mathbf{a}_i^T,\, b_i\right) : c_i < 0\right\}$.

2. Apply K-means for $\frac{K}{2}$ centers, separately for both sets of generators and receive $\left\{\tilde{c}_i\left(\tilde{\mathbf{a}}_i^T,\, \tilde{b}_i\right) : \tilde{c}_i > 0\right\}$, $\left\{\tilde{c}_i\left(\tilde{\mathbf{a}}_i^T,\, \tilde{b}_i\right) : \tilde{c}_i < 0\right\}$ as output.

3. Construct the final weights. For the first linear layer, the weights and the bias which correspond to the $i-$th neuron become the vector $\tilde{c}_i\left(\tilde{\mathbf{a}}_i^T,\, \tilde{b}_i\right)$.

4. The weights of the second linear layer are set to $1$ for every hidden layer neuron where $\tilde{c}_i\left(\tilde{\mathbf{a}}_i^T,\, \tilde{b}_i\right)$ occurs from positive generators and $-1$, elsewhere.

---

**Proposition 5.** *Zonotope K-means produces a compressed neural network with output $\tilde{v}$ satisfying*

$$\frac{1}{\rho} \cdot \max_{\mathbf{x} \in \mathcal{B}} |v(\mathbf{x}) - \tilde{v}(\mathbf{x})| \leq K \cdot \delta_{max} + \left(1 - \frac{1}{N_{max}}\right) \sum_{i=1}^{n} |c_i| \|\left(\mathbf{a}_i^T,\, b_i\right)\|$$

*where $K$ is the total number of centers used in both K-means, $\delta_{max}$ is the largest distance from a point to its corresponding cluster center and $N_{max}$ is the maximum cardinality of a cluster.*

The above proposition provides an upper bound between the original neural network and the one that is approximated with Zonotope K-means. In particular, if we use $K = n$ centers the bound of the

approximation error becomes 0, because then $\delta_{\max} = 0$ and $N_{\max} = 1$. Also, if $K \approx 0$ the bound gets a fixed value depending on the magnitude of the weights of the linear layers.

## 4.2 Multiple Zonotope Approximation

The exact positive and negative zonotope approximation performed by Zonotope K-means algorithm has a main disadvantage: it can only be used in single output neural networks. Indeed, suppose that we want to employ the preceeding algorithm to approach the zonotopes of each output in a multiclass neural network. That would require $2m$ separate executions of K-means which are not necessarily consistent. For instance, it is possible to have $c_{j_1 i} > 0$ and $c_{j_2 i} < 0$ for some output components $v_{j_1}, v_{j_2}$. That means that in the compression procedure of $v_{j_1}$, the $i-$th neuron belongs to the positive generators set, while for $v_{j_2}$, it belongs to the negative one. This makes the two compressions incompatible. Moreover, the drawback of restricting to single output only allow us to compress the final ReLU layer and not any preceeding ones.

**Neural Path K-means** To overcome this obstacle we apply a simultaneous approximation of the zonotopes. The method is called *Neural Path K-means* and directly applies K-means to the vectors of the weights $\left( \mathbf{a}_i^T, \, b_i, \, c_{1i}, \, ..., \, c_{mi} \right)$ associated to each neuron $i$ of the hidden layer. The name of the algorithm emanates from the fact that the vector associated to each neuron consists of the weights of all the neural network paths passing from this neuron. The procedure is presented in Algorithm 2.

---

**Algorithm 2:** Neural Path K-means Compression

1. Apply K-means for $K$ centers to the vectors $\left( \mathbf{a}_i^T, \, b_i, \, C_{:,i}^T \right)$, $i = 1, ..., n$, and get the centers $\left( \tilde{\mathbf{a}}_i^T, \, \tilde{b}_i, \, \tilde{C}_{:,i}^T \right)$, $i = 1, ..., K$.

2. Construct the final weights. For the first linear layer matrix the $i - th$ row becomes $\left( \tilde{\mathbf{a}}_i^T, \, \tilde{b}_i \right)$, while for the second linear layer matrix, the $i - th$ column becomes $\tilde{C}_{:,i}$.

---

**Null Generators** Neural Path K-means does not apply compression directly to each zonotope of the network, but is rather a heuristic approach for this task. More precisely, if we focus on the set of generators of the zonotopes of output $j$, Neural Path K-means might mix positive and negative generators together in the same cluster. For instance, suppose $\left( \tilde{\mathbf{a}}_k^T, \, \tilde{b}_k, \, \tilde{C}_{:,k}^T \right)$ is the cluster center corresponding to vectors $\left( \mathbf{a}_i^T, \, b_i, \, C_{:,i}^T \right)$ for $i \in I$. Then regarding output $j$, it is not necessary that $\forall i \in I$ all $c_{ji}$ have the same sign. Thus, the compressed positive zonotope $\tilde{P}_j$ might contain generators of the original negative zonotope $Q_j$ and vice versa. We will call generators $c_{ji} \left( \mathbf{a}_i^T, \, b_i \right)$ contributing to opposite zonotopes, *null generators*.

**Proposition 6.** *Neural Path K-means produces a compressed neural network with output $\tilde{v}$ satisfying*

$$\frac{1}{\rho} \cdot \max_{\mathbf{x} \in \mathcal{B}} \| v(\mathbf{x}) - \tilde{v}(\mathbf{x}) \|_1 \leq \sqrt{m} K \delta_{max}^2 + \sqrt{m} \left( 1 - \frac{1}{N_{max}} \right) \sum_{i=1}^{n} \| C_{:,i} \| \, \| \left( \mathbf{a}_i^T, \, b_i \right) \| +$$

$$\frac{\sqrt{m} \delta_{max}}{N_{min}} \sum_{i=1}^{n} \left( \| \left( \mathbf{a}_i^T, \, b_i \right) \| + \| C_{:,i} \| \right) + \sum_{j=1}^{m} \sum_{i \in \mathcal{N}_j} | c_{ji} | \, \| \left( \mathbf{a}_i^T, \, b_i \right) \|$$

*where $K$ is the number of K-means clusters, $\delta_{max}$ the maximum distance from any point to its corresponding cluster center, $N_{max}, N_{min}$ the maximum and minimum cardinality respectively of a cluster and $\mathcal{N}_j$ the set of null generators with respect to output $j$.*

The performance of Neural Path K-means is evaluated with Proposition 6. The result we deduce is analogous to Zonotope K-means. The bound of the approximation error becomes zero when $K$ approaches $n$. Indeed, for $K = n$ we get $\delta_{\max} = 0$, $N_{\max} = 1$ and $\mathcal{N}_j = \emptyset$, $\forall j \in [m]$. For lower values of $K$, the upper bound reaches a value depending on the magnitude of the weights of the linear layers together with weights corresponding to null generators.

Table 1: Reporting accuracy of compressed networks for single output compression methods.

| Percentage of remaining neurons | MNIST 3/5 | | | MNIST 4/9 | | |
|---|---|---|---|---|---|---|
| | (Smyrnis et al., 2020) | Zonotope K-means | Neural Path K-means | (Smyrnis et al., 2020) | Zonotope K-means | Neural Path K-means |
| 100% (Original) | $99.18 \pm 0.27$ | $99.38 \pm 0.09$ | $99.38 \pm 0.09$ | $99.53 \pm 0.09$ | $99.53 \pm 0.09$ | $99.53 \pm 0.09$ |
| 5% | $99.12 \pm 0.37$ | $99.42 \pm 0.07$ | $99.25 \pm 0.04$ | $98.99 \pm 0.09$ | $99.52 \pm 0.09$ | $99.48 \pm 0.15$ |
| 1% | $99.11 \pm 0.36$ | $99.39 \pm 0.05$ | $99.32 \pm 0.03$ | $99.01 \pm 0.09$ | $99.46 \pm 0.05$ | $99.35 \pm 0.17$ |
| 0.5% | $99.18 \pm 0.36$ | $99.41 \pm 0.05$ | $99.22 \pm 0.11$ | $98.81 \pm 0.09$ | $99.35 \pm 0.24$ | $98.84 \pm 1.18$ |
| 0.3% | $99.18 \pm 0.36$ | $99.25 \pm 0.37$ | $99.19 \pm 0.41$ | $98.81 \pm 0.09$ | $98.22 \pm 1.38$ | $98.22 \pm 1.33$ |

Table 2: Reporting theoretical upper bounds of Propositions 5, 6.

| Percentage of remaining neurons | MNIST 3/5 | | MNIST 4/9 | |
|---|---|---|---|---|
| | Zonotope K-means | Neural Path K-means | Zonotope K-means | Neural Path K-means |
| 100% | 0.00 | 0.00 | 0.00 | 0.00 |
| 10% | 17.07 | 246.74 | 18.85 | 229.37 |
| 2.5% | 15.35 | 59.42 | 17.02 | 63.37 |
| 1% | 14.79 | 42.22 | 16.44 | 45.58 |
| 0.5% | 14.57 | 36.47 | 16.20 | 39.71 |

## 5 EXPERIMENTS

We conduct experiments on compressing the linear layers of convolutional neural networks. Our experiments serve as proof-of-concept and indicate that our theoretical claims indeed hold in practice. The heart of our contribution lies in presenting novel tropical geometrical background for neural network approximation that will shed light for further research towards tropical mathematics.

Our methods compress the linear layers of the network layer by layer. They perform a functional approximation of the original network and thus they are applicable for both classification and regression tasks. To compare them with other techniques in the literature we choose methods with similar structure, i.e. structured pruning techniques without re-training. For example, Alfarra et al. (2020) proposed a compression algorithm based on the sparsification of the matrices representing the zonotopes which served as an intuition for part of our work. However, their method is unstructured and incompatible for comparison. The methods we choose to compare are two tropical methods for single-output (Smyrnis et al., 2020) and multi-output (Smyrnis & Maragos, 2020) networks, Random and $L1$ Structured, and a modification of ThiNet (Luo et al., 2017) adapted to linear layers. Smyrnis et al. (2020); Smyrnis & Maragos (2020) proposed a novel tropical division framework that aimed on the reduction of zonotope vertices. Random method prunes neurons according to uniform probability, while $L1$ prunes those with the lowest value of $L1$ norm of their weights. Also, ThiNet uses a greedy criterion for discarding the neurons that have the smallest contribution to the output of the network.

**MNIST Dataset, Pairs 3-5 and 4-9**  The first experiment is performed on the binary classification tasks of pairs $3/5$ and $4/9$ of the MNIST dataset and so we can utilize both of our proposed methods. In Table 1, we compare our methods with a tropical geometrical approach of Smyrnis et al. (2020). Their method is based on a tropical division framework for network minimization. For fair comparison, we use the same CNN with two fully connected layers and hidden layer of size 1000. According to Table 1, our techniques seem to have similar performance. They retain the accuracy of the network while reducing its size. Moreover, in Table 2 we include experimental computation of the theoretical bounds provided by Proposition 5, 6. We notice that the bounds decrease as the remaining weights get less. The behaviour of the bounds was expected to be incremental because the less weights we use, the compression gets worse and the error becomes larger. However, the opposite holds which means that the bounds are tighter for higher pruning rates. It is also important to mention that the bounds become 0 when we keep all the weights, as expected.

**MNIST and Fashion-MNIST Datasets**  For the second experiment we employ MNIST and Fashion-MNIST datasets. The corresponding classification is multiclass and thus Neural Path K-means may only be applied. In Table 3, we compare it with the multiclass tropical method of Smyrnis & Maragos (2020) using the same CNN architecture they do. Furthermore, in plots 4a, 4b we compare Neural Path K-means with ThiNet and baseline pruning methods by compressing

Table 3: Reporting accuracy of compressed networks for multiclass compression methods.

| Percentage of remaining neurons | MNIST | | Fashion-MNIST | |
|---|---|---|---|---|
| | (Smyrnis & Maragos, 2020) | Neural Path K-means | (Smyrnis & Maragos, 2020) | Neural Path K-means |
| 100% (Original) | $98.60 \pm 0.03$ | $98.61 \pm 0.11$ | $88.66 \pm 0.54$ | $89.52 \pm 0.19$ |
| 50% | $96.39 \pm 1.18$ | $98.13 \pm 0.28$ | $83.30 \pm 2.80$ | $88.22 \pm 0.32$ |
| 25% | $95.15 \pm 2.36$ | $98.42 \pm 0.42$ | $82.22 \pm 2.85$ | $86.67 \pm 1.12$ |
| 10% | $93.48 \pm 2.57$ | $96.89 \pm 0.55$ | $80.43 \pm 3.27$ | $86.04 \pm 0.94$ |
| 5% | $92.93 \pm 2.59$ | $96.31 \pm 1.29$ | $-$ | $83.68 \pm 1.06$ |

LeNet5 (LeCun et al., 1998). To get a better idea of how our method performs in deeper architectures we provide plots 4c,4d that illustrate the performance of compressing a deep neural network with layers of size $28 * 28, 512, 256, 128$ and $10$, which we refer to as deepNN. The compression is executed on all hidden layers beginning from the input and heading to the output. From Table 3, we deduce that our method performs better than (Smyrnis & Maragos, 2020). Also, it achieves higher accuracy scores and experience lower variance as shown in plots 4a-4d. Neural Path K-means, overall, seems to have good performance, even competitive to ThiNet. Its worst performance occurs on low percentages of remaining weights. An explanation for this is that K-means provides a high-quality compression as long as the number of centers is not less than the number of "real" clusters.

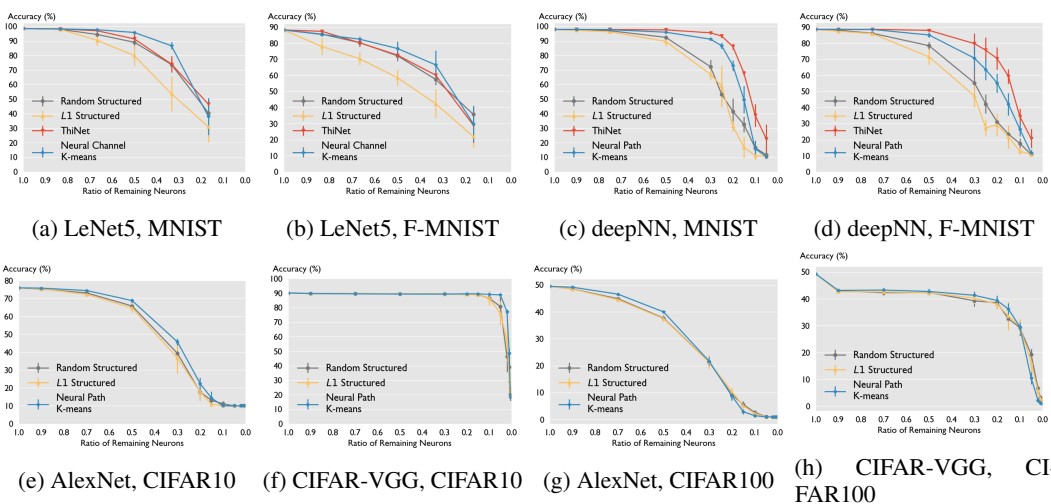

Figure 4: Neural Path K-means compared with baseline pruning methods and ThiNet. Horizontal axis shows the ratio of remaining neurons in each hidden layer of the fully connected part.

**CIFAR Dataset** We conduct our final experiment on CIFAR datasets using CIFAR-VGG (Blalock et al., 2020) and an altered version of AlexNet adapted for CIFAR. The resulting plots are shown in Fig. 4e-4h. We deduce that Neural Path K-means retains a good performance on larger datasets. In particular, in most cases it has slightly better accuracy an lower deviation than the baselines, but has worse behaviour when keeping almost zero weights.

# 6    CONCLUSIONS AND FUTURE WORK

We presented a novel theorem on the bounding of the approximation error between two neural networks. This theorem occurs from the bounding of the tropical polynomials representing the neural networks via the Hausdorff distance of their extended Newton polytopes. We derived geometrical compression algorithms for the fully connected parts of ReLU activated deep neural networks, while application to convolutional layers is an ongoing work. Our algorithms seem to perform well in practice and motivate further research towards the direction revealed by tropical geometry.

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

## A PROOFS FOR THE SECTION "BACKGROUND ON TROPICAL GEOMETRY"

### A.1 PROOF OF PROPOSITION 3

*Proof.* The first argument follows from the fact that both $p_j, q_j$ are linear combinations of tropical polynomials consisting of two terms. Indeed, we compute

$$p_j(\mathbf{x}) = \sum_{c_{ji}>0} c_{ji} \max(\mathbf{a}_i^T \mathbf{x} + b_i, 0) = \sum_{c_{ji}>0} \max(c_{ji}\mathbf{a}_i^T \mathbf{x} + c_{ji}b_i, 0) \xRightarrow{\text{Prop. 1}}$$

$$P_j = \bigoplus_{c_{ji}>0} \text{ENewt}\left(\max(c_{ji}\mathbf{a}_i^T \mathbf{x} + c_{ji}b_i, 0)\right)$$

Each $\text{ENewt}\left(\max(c_{ji}\mathbf{a}_i^T \mathbf{x} + c_{ji}b_i, 0)\right)$ is a line segment with endpoints $\mathbf{0}$ and $\left(c_{ji}\mathbf{a}_i^T, c_{ji}b_i\right) = c_{ji}\left(\mathbf{a}_i^T, b_i\right)$. Therefore $P_j$ is written as the Minkowski sum of line segments, which is a zonotope by definition. Similarly $Q_j$ is a zonotope.

Furthermore, from the definition of the Minkowski sum, each point $\mathbf{v} \in P_j$ may be written as $\sum_{c_{ji}>0} \mathbf{v}_i$, where each $\mathbf{v}_i$ is a point in the segment $\text{ENewt}\left(\max(c_{ji}\mathbf{a}_i^T \mathbf{x} + c_{ji}b_i, 0)\right)$. A vertex of $P_j$ can only occur if $\mathbf{v}_i$ is an extreme point of $\text{ENewt}\left(\max(c_{ji}\mathbf{a}_i^T \mathbf{x} + c_{ji}b_i, 0)\right)$ for every $i$ which is equivalent to either $\mathbf{v}_i = \mathbf{0}$ or $\mathbf{v}_i = c_{ji}\left(\mathbf{a}_i^T, b_i\right)$. This means that every vertex of $P_j$ corresponds to a subset $I_+ \subseteq [n]$ of indices $i$ with $c_{ji} > 0$, for which we choose $\mathbf{v}_i = c_{ji}\left(\mathbf{a}_i^T, b_i\right)$ and for the rest it holds $\mathbf{v}_i = \mathbf{0}$. Thus,

$$\mathbf{v} = \sum_{i \in I_+} c_{ji}\left(\mathbf{a}_i^T, b_i\right)$$

In the same way we derive the analogous result for the negative zonotope $Q_j$. $\square$

**Corollary 2.** *The geometric result concerning the structure of zonotopes can be extended to max-pooling layers. For instance, a max-pooling layer of size $2 \times 2$ corresponds to a polytope that is constructed as the Minkowski sum of pyramids which could stand as generalized case of zonotope.*

## B PROOFS FOR THE SECTION "APPROXIMATION OF TROPICAL POLYNOMIALS"

### B.1 PROOF OF PROPOSITION 4

*Proof.* Consider a point $\mathbf{x} \in \mathcal{B}$ and assume that $p(\mathbf{x}) = \mathbf{a}^T \mathbf{x} + b$, $\tilde{p}(\mathbf{x}) = \mathbf{c}^T \mathbf{x} + d$. Then,

$$p(\mathbf{x}) - \tilde{p}(\mathbf{x}) = p(\mathbf{x}) - \max_{(\tilde{\mathbf{a}}^T, \tilde{b}) \in \mathcal{V}_{\tilde{p}}} \{\tilde{\mathbf{a}}^T \mathbf{x} + \tilde{b}\} \leq \mathbf{a}^T \mathbf{x} + b - \left(\mathbf{u}^T, v\right)\begin{pmatrix}\mathbf{x}\\1\end{pmatrix}$$

where $\left(\mathbf{u}^T, v\right)$ may be any vertex of $\tilde{P}$. Similarly, we derive

$$\left(\mathbf{r}^T, s\right)\begin{pmatrix}\mathbf{x}\\1\end{pmatrix} - \left(\mathbf{c}^T \mathbf{x} + d\right) \leq p(\mathbf{x}) - \tilde{p}(\mathbf{x})$$

for any vertex $\left(\mathbf{r}^T, s\right)$ of $P$. Therefore, we may select the vertices $\left(\mathbf{u}^T, v\right) \in \tilde{P}$, $\left(\mathbf{r}^T, s\right) \in P$ so that their respective distances from $\left(\mathbf{a}^T, b\right)$ and $\left(\mathbf{c}^T, d\right)$, respectively, are minimized. Choosing them in such a way gives

$$p(\mathbf{x}) - \tilde{p}(\mathbf{x}) \leq \mathbf{a}^T \mathbf{x} + b - \left(\mathbf{u}^T, v\right)\begin{pmatrix}\mathbf{x}\\1\end{pmatrix} = \left(\left(\mathbf{a}^T, b\right) - \left(\mathbf{u}^T, v\right)\right)\begin{pmatrix}\mathbf{x}\\1\end{pmatrix} \leq$$

$$\leq \left\|\left(\mathbf{a}^T, b\right) - \left(\mathbf{u}^T, v\right)\right\| \left\|\begin{pmatrix}\mathbf{x}\\1\end{pmatrix}\right\| \leq d\left(\left(\mathbf{a}^T, b\right), \tilde{P}\right)\sqrt{r^2 + 1} \tag{5}$$

In similar manner we deduce

$$p(\mathbf{x}) - \tilde{p}(\mathbf{x}) \geq \left(\mathbf{r}^T, s\right)\begin{pmatrix}\mathbf{x}\\1\end{pmatrix} - \mathbf{c}^T \mathbf{x} + d = \left(\left(\mathbf{r}^T, s\right) - \left(\mathbf{c}^T, d\right)\right)\begin{pmatrix}\mathbf{x}\\1\end{pmatrix} \geq$$

$$\geq -\left\|\left(\mathbf{r}^T, s\right) - \left(\mathbf{c}^T, d\right)\right\| \left\|\begin{pmatrix}\mathbf{x}\\1\end{pmatrix}\right\| \geq -d\left(P, \left(\mathbf{c}^T, d\right)\right)\sqrt{r^2 + 1} \tag{6}$$

Notice, that for the relations (5) and (6) we used Cauchy-Schwartz inequality

$$|\langle \mathbf{x}, \mathbf{y} \rangle| \leq \|\mathbf{x}\| \|\mathbf{y}\| \Leftrightarrow -\|\mathbf{x}\| \|\mathbf{y}\| \leq \langle \mathbf{x}, \mathbf{y} \rangle \leq \|\mathbf{x}\| \|\mathbf{y}\|$$

Inequality (5) holds at any point $x \in \mathcal{B}$ for some vertex $(a^T, b) \in P$, therefore

$$p(\mathbf{x}) - \tilde{p}(\mathbf{x}) \leq \rho \cdot \max_{(\mathbf{a}^T, b) \in \mathcal{V}_P} d\left(\left(\mathbf{a}^T, b\right), \mathcal{V}_{\tilde{P}}\right) \tag{7}$$

for all $x \in \mathcal{B}$. Similarly, we derive

$$p(\mathbf{x}) - \tilde{p}(\mathbf{x}) \geq \min_{(\mathbf{c}, d) \in \mathcal{V}_{\tilde{P}}} -\rho \cdot d\left(\mathcal{V}_P, \left(\mathbf{c}^T, d\right)\right) = -\max_{(\mathbf{c}^T, d) \in \mathcal{V}_{\tilde{P}}} \rho \cdot d\left(\mathcal{V}_P, \left(\mathbf{c}^T, d\right)\right) \tag{8}$$

Combining (7) and (8) gives

$$-\max_{(\mathbf{c}^T, d) \in \mathcal{V}_{\tilde{P}}} \rho \cdot d\left(\mathcal{V}_P, \left(\mathbf{c}^T, d\right)\right) \leq p(\mathbf{x}) - \tilde{p}(\mathbf{x}) \leq \max_{(\mathbf{a}^T, b) \in \mathcal{V}_P} \rho \cdot d\left(\left(\mathbf{a}^T, b\right), \mathcal{V}_{\tilde{P}}\right) \Leftrightarrow$$

$$|p(\mathbf{x}) - \tilde{p}(\mathbf{x})| \leq \rho \cdot \max \left\{ \max_{(\mathbf{a}^T, b) \in \mathcal{V}_P} \rho \cdot d\left(\left(\mathbf{a}^T, b\right), \mathcal{V}_{\tilde{P}}\right), \max_{(\mathbf{c}^T, d) \in \mathcal{V}_{\tilde{P}}} \rho \cdot d\left(\mathcal{V}_P, \left(\mathbf{c}^T, d\right)\right) \right\}$$

Hence, from the definition of the Hausdorff distance of two polytopes we derive the desired upper bound

$$|p(\mathbf{x}) - \tilde{p}(\mathbf{x})| \leq \rho \cdot \mathcal{H}\left(P, \tilde{P}\right), \forall \mathbf{x} \in \mathcal{B} \Rightarrow$$

$$\max_{\mathbf{x} \in \mathcal{B}} |p(\mathbf{x}) - \tilde{p}(\mathbf{x})| \leq \rho \cdot \mathcal{H}\left(P, \tilde{P}\right)$$

$$\square$$

**Remark 2.** Note that with similar proof one may replace the Hausdorff distance of the two polytopes by the Hausdorff distance of their upper envelopes. This makes our theorem an exact generalization of Proposition 2. However, this format is difficult to use in practice, because it is computationally harder to determine the vertices of the upper envelope.

### B.2 Proof of Theorem 1

*Proof.* Notice that we may write

$$\|v(\mathbf{x}) - \tilde{v}(\mathbf{x})\|_1 = \sum_{j=1}^m |v_j(\mathbf{x}) - \tilde{v}_j(\mathbf{x})| = \sum_{j=1}^m |(p_j(\mathbf{x}) - q_j(\mathbf{x})) - (\tilde{p}_j(\mathbf{x}) - \tilde{q}_j(\mathbf{x}))|$$

$$= \sum_{j=1}^m |(p_j(\mathbf{x}) - \tilde{p}_j(\mathbf{x})) - (q_j(\mathbf{x}) - \tilde{q}_j(\mathbf{x}))| \leq \sum_{j=1}^m |p_j(\mathbf{x}) - \tilde{p}_j(\mathbf{x})| + |q_j(\mathbf{x}) - \tilde{q}_j(\mathbf{x})|$$

Thus from from Proposition 4 we derive

$$\max_{\mathbf{x} \in \mathcal{B}} \|v(\mathbf{x}) - \tilde{v}(\mathbf{x})\|_1 \leq \rho \cdot \left( \sum_{j=1}^m \mathcal{H}\left(P_j, \tilde{P}_j\right) + \mathcal{H}\left(Q_j, \tilde{Q}_j\right) \right)$$

$$\square$$

## C  Proofs and Figures for the section "Neural Network Compression Algorithms"

### C.1  Illustration of Zonotope K-means

Below we present a larger version of Fig. 5 demonstrating the execution of Zonotope K-means .

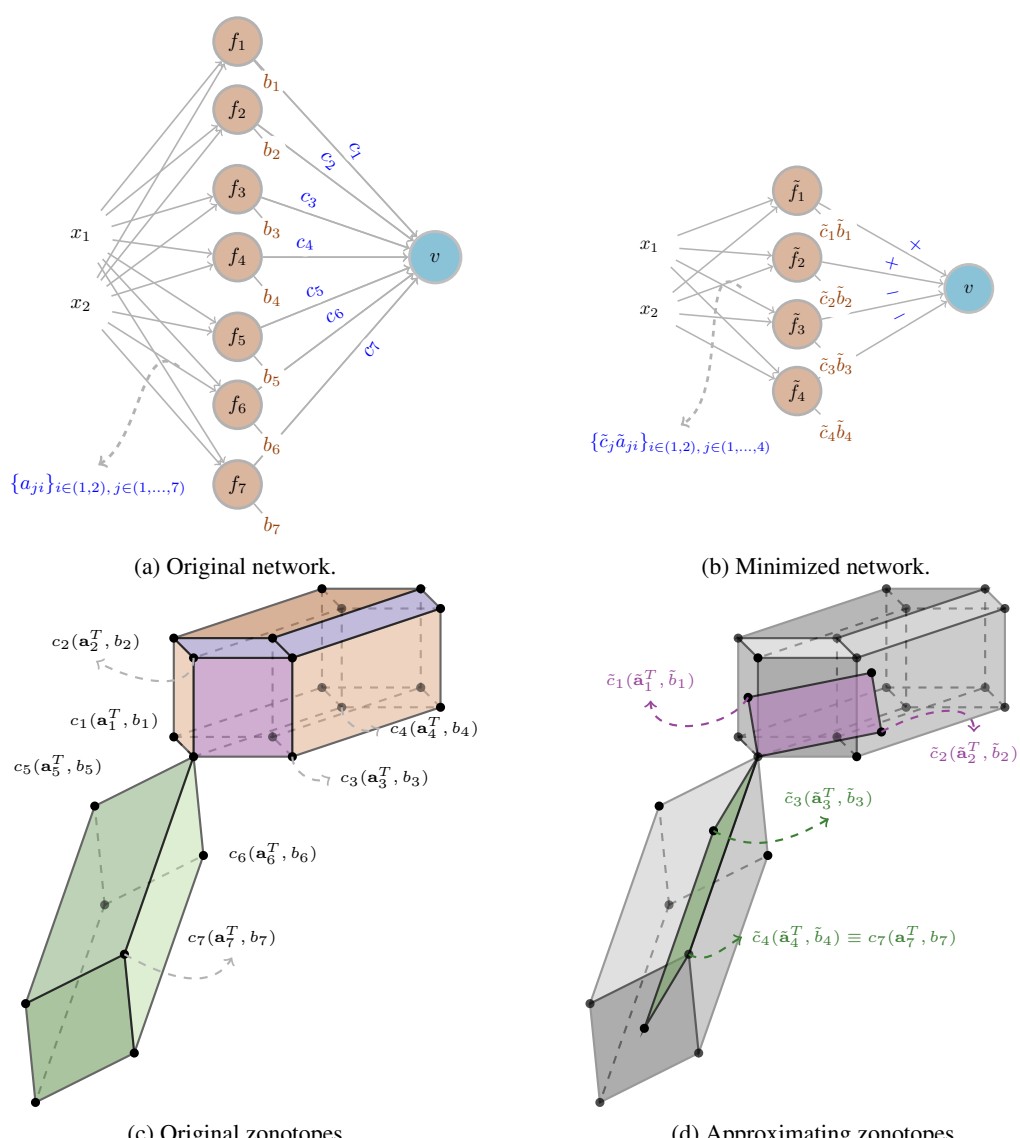

(a) Original network.

(b) Minimized network.

(c) Original zonotopes

(d) Approximating zonotopes.

Figure 5: Illustration of Zonotope K-means execution. The original zonotope $P$ is generated by $c_i \left( \mathbf{a}_i^T, b_i \right)$ for $i = 1, ..., 4$ and the negative zonotope $Q$ generated by the remaining ones $i = 5, 6, 7$. The approximation $\tilde{P}$ of $P$ is colored in purple and generated by $\tilde{c}_i \left( \tilde{\mathbf{a}}_i^T, \tilde{b}_i \right)$, $i = 1, 2$ where the first generator is the K-means center representing the generators $1, 2$ of $P$ and the second is the representative center of $3, 4$. Similarly, the approximation $\tilde{Q}$ of $Q$ is colored in green and defined by the generators $\tilde{c}_i \left( \tilde{\mathbf{a}}_i^T, \tilde{b}_i \right)$, $i = 3, 4$ that stand as representative centers for $\{5, 6\}$ and $7$ respectively.

## C.2 PROOF OF PROPOSITION 5

*Proof.* We remind that for the output functions it holds

$$v(\mathbf{x}) = p(\mathbf{x}) - q(\mathbf{x}) \ , \tilde{v}(\mathbf{x}) = \tilde{p}(\mathbf{x}) - \tilde{q}(\mathbf{x})$$

From triangle inequality we deduce

$$|v(\mathbf{x}) - \tilde{v}(\mathbf{x})| = |p(\mathbf{x}) - q(\mathbf{x}) - (\tilde{p}(\mathbf{x}) - \tilde{q}(\mathbf{x}))| < |p(\mathbf{x}) - \tilde{p}(\mathbf{x})| + |q(\mathbf{x}) - \tilde{q}(\mathbf{x})|$$

Prop 4 bounds $|p(\mathbf{x}) - \tilde{p}(\mathbf{x})|$ and $|q(\mathbf{x}) - \tilde{q}(\mathbf{x})|$ are bounded by $\mathcal{H}\left( P, \tilde{P} \right)$ and $\mathcal{H}\left( Q, \tilde{Q} \right)$ respectively. Therefore, it suffices to get an upper bound for these Hausdorff distances. Let us consider any

vertex $\mathbf{u} = \sum_{i \in I_+} c_i \left( \mathbf{a}_i^T, \, b_i \right)$ of $P$. For the vertex $\mathbf{u} \in P$ we need to choose vertex $\mathbf{v} \in \tilde{P}$ as close to $\mathbf{u}$ as possible, in order to provide an upper bound for $\mathrm{dist} \left( \mathbf{u}, \tilde{P} \right)$. Vertex $\mathbf{v}$ is selected as follows. For each $i \in I_+$ we select $k$ such that $\tilde{c}_k \begin{pmatrix} \tilde{\mathbf{a}}_k^T & \tilde{b}_k \end{pmatrix}$ is the center of the cluster where $c_i \left( \mathbf{a}_i^T, \, b_i \right)$ belongs to. We denote the set of such clusters by $C_+$, where each cluster center $k$ appears only once. Then, vertex $\mathbf{v}$ is constructed as $\mathbf{v} = \sum_{k \in C_+} \tilde{c}_k \begin{pmatrix} \tilde{\mathbf{a}}_k^T & \tilde{b}_k \end{pmatrix} \in \tilde{P}$. We have that:

$$
\begin{aligned}
\mathrm{dist} \left( \mathbf{u}, \tilde{P} \right) &\le \left\| \sum_{i \in I_+} c_i \left( \mathbf{a}_i^T, \, b_i \right) - \sum_{k \in C_+} \tilde{c}_k \left( \tilde{\mathbf{a}}_k^T, \, \tilde{b}_k \right) \right\| \\
&\le \sum_{k \in C_+} \left\| \sum_{i \in I_{k_+}} c_i \left( \mathbf{a}_i^T, \, b_i \right) - \tilde{c}_k \left( \tilde{\mathbf{a}}_k^T, \, \tilde{b}_k \right) \right\| \\
&\le \sum_{k \in C_+} \sum_{i \in I_{k_+}} \left\| c_i \left( \mathbf{a}_i^T, \, b_i \right) - \frac{\tilde{c}_k \left( \tilde{\mathbf{a}}_k^T, \, \tilde{b}_k \right)}{|I_{k_+}|} \right\| \\
&= \sum_{k \in C_+} \sum_{i \in I_{k_+}} \left\| c_i \left( \mathbf{a}_i^T, \, b_i \right) - \frac{c_i \left( \mathbf{a}_i^T, \, b_i \right) + \varepsilon_i}{|I_{k_+}|} \right\| \\
&\le \sum_{k \in C_+} \sum_{i \in I_{k_+}} \left[ \left( 1 - \frac{1}{|I_{k_+}|} \right) |c_i| \left\| \left( \mathbf{a}_i^T, \, b_i \right) \right\| + \frac{\|\varepsilon_i\|}{|I_{k_+}|} \right] \\
&\le |C_+| \cdot \delta_{\max} + \left( 1 - \frac{1}{N_{\max}} \right) \sum_{i \in I_+} |c_i| \left\| \left( \mathbf{a}_i^T, \, b_i \right) \right\|
\end{aligned}
$$

where we denote by $I_{k_+}$ the set of indices $i \in I_+$ that belong to the center $k \in C_+$ and $\varepsilon_i = \tilde{c}_k \left( \tilde{\mathbf{a}}_k^T, \, \tilde{b}_k \right) - c_i \left( \mathbf{a}_i^T, \, b_i \right)$ is the vector of the difference of the $i-$th generator, with its corresponding K-means cluster center.

The maximum value of the upper bound occurs when $I_+$ contains all indices that correspond to $c_i > 0$. This value gives us an upper bound for $\max_{\mathbf{u} \in P} d(\mathbf{u}, \tilde{P})$. To compute an upper bound for $\max_{\mathbf{v} \in \mathcal{V}_{\tilde{P}}} d(P, \mathbf{v})$ we assume $\mathbf{v} = \sum_{k \in C_+} \tilde{c}_k \begin{pmatrix} \tilde{\mathbf{a}}_k^T & \tilde{b}_k \end{pmatrix}$ and consider the vertex $\sum_{i \in I_+} c_i \left( \mathbf{a}_i^T, \, b_i \right) \in P$ where $I_+$ is the set of indices of positive generators corresponding to the union of all clusters corresponding to the centers of $C_+$. Note that the occurring distance

$$
\left\| \sum_{i \in I_+} c_i \left( \mathbf{a}_i^T, \, b_i \right) - \sum_{k \in C_+} \tilde{c}_k \left( \tilde{\mathbf{a}}_i^T, \, \tilde{b}_i \right) \right\|
$$

was taken into account when computing the upper bound for $\max_{\mathbf{u} \in \mathcal{V}_P} d(\mathbf{u}, \tilde{P})$, and thus both values obtain the same upper bound. Therefore,

$$
\mathcal{H} \left( P, \tilde{P} \right) \le K_+ \cdot \delta_{\max} + \left( 1 - \frac{1}{N_{\max}} \right) \sum_{i \in I_+} |c_i| \left\| \left( \mathbf{a}_i^T, \, b_i \right) \right\|
$$

where $K_+$ is the number of cluster centers corresponding to $\tilde{P}$ and $I_+$ the indices corresponding to all positive generators of $P$. Similarly,

$$
\mathcal{H} \left( Q, \tilde{Q} \right) \le K_- \cdot \delta_{\max} + \left( 1 - \frac{1}{N_{\max}} \right) \sum_{i \in I_-} |c_i| \left\| \left( \mathbf{a}_i^T, \, b_i \right) \right\|
$$

where $K_-, I_-$ are defined in analogous way for the negative zonotope. Combining the relations gives the desired bound.

$$
\frac{1}{\rho} \cdot |v(\mathbf{x}) - \tilde{v}(\mathbf{x})| \le \mathcal{H} \left( P, \tilde{P} \right) + \mathcal{H} \left( Q, \tilde{Q} \right) \le K \cdot \delta_{\max} + \left( 1 - \frac{1}{N_{\max}} \right) \sum_{i=1}^n |c_i| \left\| \left( \mathbf{a}_i^T, \, b_i \right) \right\|
$$

$\square$

### C.3 ILLUSTRATION FOR NEURAL PATH K-MEANS ALGORITHM

Below we illustrate the vectors on which K-means is applied for the multi-output case. The vectors that are compressed are consist of all the edges associated to a hidden layer neuron. The corresponding edges contain all the possible neural paths that begin from some node of the input, end in some node of the output and pass through this hidden node.

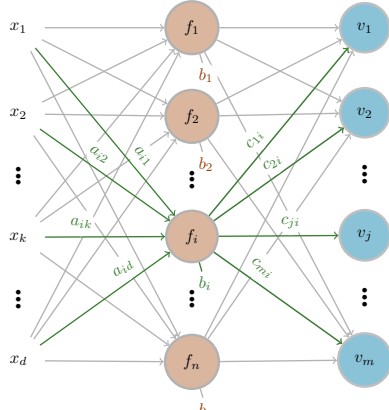

Figure 6: Neural Path K-means for multi-output neural network compression. In green color we highlight the weights corresponding to the $i-$th vector used by Neural Path K-means.

In the main text we defined the null generators of zonotopes that occur by the execution of Neural Path K-means. Below we provide an illustration for them.

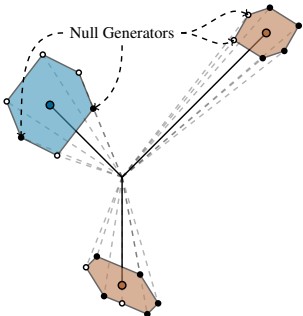

Figure 7: Visualization of K-means in $\mathbb{R}^{d+1+n}$, where $d$ is the input dimension and $n$ the hidden layer size. We color points according to the $j-$th output component of the network. Black and white points correspond to generators of $P_j$ and $Q_j$ respectively. White vertices in positive (brown) clusters and black vertices in negative (blue) clusters are null generators regarding $j-$th output.

### C.4 PROOF OF PROPOSITION 6

*Proof.* Let us first focus on a single output, say $j-$th output. As in the proof for Zonotope K-means, we will bound $\mathcal{H}\left(P_j, \tilde{P}_j\right), \mathcal{H}\left(Q_j, \tilde{Q}_j\right)$ for all $j \in [m]$. From triangle inequality we get

$$|v_j(\mathbf{x}) - \tilde{v}_j(\mathbf{x})| \leq |p_j(\mathbf{x}) - \tilde{p}_j(\mathbf{x})| + |q_j(\mathbf{x}) - \tilde{q}_j(\mathbf{x})|$$

Any vertex of $\mathbf{u} \in P_j$ can be written as $\mathbf{u} = \sum_{i \in I_{j+}} c_{ji} \left(\mathbf{a}_i^T, b_i\right)$ where the set of indices $I_{j+}$ satisfy $c_{ji} > 0, \forall i \in I_{j+}$ and thus correspond to positive generators. To choose a nearby vertex from $\tilde{P}_j$ we perform the following. For each $i \in I_{j+}$ we select the center $\left(\tilde{\mathbf{a}}_k^T \quad \tilde{b}_k \quad \tilde{C}^{(k)T}\right)$ of the cluster to where $\left(\mathbf{a}_i^T \quad b_i \quad C^{(i)T}\right)$ belongs, only if $\tilde{c}_{jk} > 0$. Such a center only exists if $c_{ji} \left(\mathbf{a}_i^T, b_i\right)$ is not a

null generator. Else, we choose as representation the vector $\mathbf{0}$. Each cluster center, or $\mathbf{0}$, is taken into account once and the vertex $\sum_{k \in C_{j+}} \tilde{c}_{jk} \left( \tilde{\mathbf{a}}_k, \tilde{b}_k \right) \in \tilde{P}_j$ is formed. We derive:

$$
\begin{aligned}
\max_{u \in \mathcal{V}_{P_j}} \text{dist} \left( u, \tilde{P}_j \right) &\leq \left\| \sum_{i \in I_{j+}} c_{ji} \left( \mathbf{a}_i^T, b_i \right) - \sum_{k \in C_{j+}} \tilde{c}_{jk} \left( \tilde{\mathbf{a}}_k^T, \tilde{b}_k \right) \right\| \\
&\leq \sum_{k \in C_{j+}} \left\| \sum_{i \in I_{jk+}} c_{ji} \left( \mathbf{a}_i^T, b_i \right) - \tilde{c}_{jk} \left( \tilde{\mathbf{a}}_k^T, \tilde{b}_k \right) \right\| + \sum_{i \in N_{j+}} |c_{ji}| \left\| \left( \mathbf{a}_i^T, b_i \right) \right\| \\
&\leq \sum_{k \in C_{j+}} \sum_{i \in I_{jk+}} \left\| c_{ji} \left( \mathbf{a}_i^T, b_i \right) - \frac{\tilde{c}_{jk} \left( \tilde{\mathbf{a}}_k^T, \tilde{b}_k \right)}{|I_{jk+}|} \right\| + \sum_{i \in N_{j+}} |c_{ji}| \left\| \left( \mathbf{a}_i^T, b_i \right) \right\| \\
&\leq \sum_{k \in C} \sum_{i \in I_{jk+}} \left\| c_{ji} \left( \mathbf{a}_i^T, b_i \right) - \frac{(c_{ji} + \varepsilon_{ji}) \left[ \left( \mathbf{a}_i^T, b_i \right) + \lambda_i \right]}{|I_{jk+}|} \right\| + \sum_{i \in N_{j+}} |c_{ji}| \left\| \left( \mathbf{a}_i^T, b_i \right) \right\| \\
&\leq \sum_{k \in C_{j+}} \sum_{i \in I_{jk+}} \left[ \frac{|\varepsilon_{ji}| \|\lambda_i\|}{|I_{jk+}|} + \left( 1 - \frac{1}{|I_{jk+}|} \right) |c_{ji}| \left\| \left( \mathbf{a}_i^T, b_i \right) \right\| \right] + \\
&+ \sum_{k \in C_{j+}} \sum_{i \in I_{jk+}} \left[ \frac{|\varepsilon_{ji}| \left\| \left( \mathbf{a}_i^T, b_i \right) \right\| + |c_{ji}| \|\lambda_i\|}{|I_{jk+}|} \right] + \sum_{i \in N_{j+}} |c_{ji}| \left\| \left( \mathbf{a}_i^T, b_i \right) \right\|
\end{aligned}
$$

where for all $i \in I_{jk+}$ the $i-$th vector of K-means is represented by the $k-$th center $k \in C_{j+}$. We also assumed $\tilde{C}^{(i)} = C_{:,i} + \varepsilon^{(i)} \Rightarrow \tilde{c}_{ji} = c_{ji} + \varepsilon_{ji}$ and $\left( \tilde{\mathbf{a}}_i^T, \tilde{b}_i \right) = \left( \mathbf{a}_i^T, b_i \right) + \lambda_i$.

The maximum value of the upper bound occurs when $I_{j+}$ contains all indices that correspond to $c_{ji} > 0$. To compute an upper bound for $\max_{\mathbf{v} \in \mathcal{V}_{\tilde{P}_j}} \text{dist} \left( P_j, \mathbf{v} \right)$ we write the vertex $\mathbf{v}$ as $\mathbf{v} = \sum_{k \in C_{j+}} \tilde{c}_{jk} \left( \tilde{\mathbf{a}}_k^T \quad \tilde{b}_k \right) \in \tilde{P}_j$ and choose the vertex $\mathbf{u} = \sum_{i \in I_{j+}} c_{ji} \left( \mathbf{a}_i^T, b_i \right)$ of $P_j$ where $I_{j+}$ is the set of all indices corresponding to generators that belong to these clusters. As in the proof of Proposition 5, their distance was taken into account when computing the upper bound for $\max_{\mathbf{u} \in \mathcal{V}_{P_j}} \text{dist} \left( \mathbf{u}, \tilde{P}_j \right)$. Hence, both obtain the same upper bound which leads to

$$
\begin{aligned}
\mathcal{H} \left( P_j, \tilde{P}_j \right) &\leq \sum_{k \in C_{j+}} \sum_{i \in I_{jk+}} \left[ \frac{|\varepsilon_{ji}| \|\lambda_i\|}{|I_{jk+}|} + \left( 1 - \frac{1}{|I_{jk+}|} \right) |c_{ji}| \left\| \left( \mathbf{a}_i^T, b_i \right) \right\| \right] + \\
&+ \sum_{k \in C_{j+}} \sum_{i \in I_{jk+}} \left[ \frac{|\varepsilon_{ji}| \left\| \left( \mathbf{a}_i^T, b_i \right) \right\| + |c_{ji}| \|\lambda_i\|}{|I_{jk+}|} \right] + \sum_{i \in N_{j+}} |c_{ji}| \left\| \left( \mathbf{a}_i^T, b_i \right) \right\|
\end{aligned}
$$

where $I_{j+}$ contains all indices corresponding to positive $c_{ji}$. Similarly, we deduce

$$
\begin{aligned}
\mathcal{H} \left( Q_j, \tilde{Q}_j \right) &\leq \sum_{k \in C_{j-}} \sum_{i \in I_{jk-}} \left[ \frac{|\varepsilon_{ji}| \|\lambda_i\|}{|I_{jk-}|} + \left( 1 - \frac{1}{|I_{jk-}|} \right) |c_{ji}| \left\| \left( \mathbf{a}_i^T, b_i \right) \right\| \right] + \\
&+ \sum_{k \in C_{j-}} \sum_{i \in I_{jk-}} \left[ \frac{|\varepsilon_{ji}| \left\| \left( \mathbf{a}_i^T, b_i \right) \right\| |c_{ji}| \|\lambda_i\|}{|I_{jk-}|} \right] + \sum_{i \in N_{j-}} |c_{ji}| \left\| \left( \mathbf{a}_i^T, b_i \right) \right\|
\end{aligned}
$$

where $I_{j-}$ contains all $i$ such that $c_{ji} < 0$. In total these bounds together with Proposition 4 give

$$\frac{1}{\rho} \cdot \max_{\mathbf{x} \in \mathcal{B}} |v_j(\mathbf{x}) - \tilde{v}_j(\mathbf{x})| \le \sum_{k \in C_j} \sum_{i \in I_{jk}} \left[ \frac{|\varepsilon_{ji}| \|\lambda_i\|}{|I_{jk}|} + \left( 1 - \frac{1}{|I_{jk}|} \right) |c_{ji}| \left\| \left( \mathbf{a}_i^T, b_i \right) \right\| \right] +$$

$$+ \sum_{k \in C_j} \sum_{i \in I_{jk}} \left[ \frac{|\varepsilon_{ji}| \left\| \left( \mathbf{a}_i^T, b_i \right) \right\| + |c_{ji}| \|\lambda_i\|}{|I_{jk}|} \right] + \sum_{i \in N_j} |c_{ji}| \left\| \left( \mathbf{a}_i^T, b_i \right) \right\|$$

Here we used the notation $C_j = C_{j+} \cup C_{j-} = \{1, 2, ..., K\}$ and $I_{jk}$ is either equal to $I_{jk+}$ or $I_{jk-}$ depending on $k \in C_j$. Note that $\{i | i \in I_{jk}, k \in C_j\} = \{1, 2, ..., n\} \setminus N_j \subseteq \{1, 2, ..., n\}$, since every generator that is not null corresponds to some cluster center with the same sign. Also, using $N_{\max} \ge |I_{jk}| \ge N_{\min}$, it follows that

$$\frac{1}{\rho} \cdot \max_{\mathbf{x} \in \mathcal{B}} |v_j(\mathbf{x}) - \tilde{v}_j(\mathbf{x})| \le \sum_{i=1}^{n} \left[ \frac{|\varepsilon_{ji}| \|\lambda_i\|}{N_{\min}} + \left( 1 - \frac{1}{N_{\max}} \right) |c_{ji}| \left\| \left( \mathbf{a}_i^T, b_i \right) \right\| \right] +$$

$$+ \sum_{i=1}^{n} \left[ \frac{|\varepsilon_{ji}| \left\| \left( \mathbf{a}_i^T, b_i \right) \right\| + |c_{ji}| \|\lambda_i\|}{N_{\min}} \right] + \sum_{i \in N_j} |c_{ji}| \left\| \left( \mathbf{a}_i^T, b_i \right) \right\|$$

We will compute the total cost that combines all outputs by applying the inequality

$$\left( \sum_{j=1}^{m} |u_j| \right)^2 \le m \left( \sum_{j=1}^{m} |u_j|^2 \right) \Leftrightarrow \sum_{j=1}^{m} |u_j| \le \sqrt{m} \, \|(u_1, ..., u_m)\|$$

which is a direct application of Cauchy-Schwartz inequality. Together with the relations $\|\varepsilon^{(i)}\| < \delta_{\max}, \quad \|\lambda_i\| < \delta_{\max}$, we get

$$\frac{1}{\rho} \cdot \sum_{j=1}^{m} \max_{\mathbf{x} \in \mathcal{B}} |v_j(\mathbf{x}) - \tilde{v}_j(\mathbf{x})| \le \sum_{j=1}^{m} \sum_{i=1}^{n} \left[ \frac{|\varepsilon_{ji}| \|\lambda_i\|}{N_{\min}} + \left( 1 - \frac{1}{N_{\max}} \right) |c_{ji}| \left\| \left( \mathbf{a}_i^T, b_i \right) \right\| \right] +$$

$$+ \sum_{j=1}^{m} \sum_{i=1}^{n} \left[ \frac{|\varepsilon_{ji}| \left\| \left( \mathbf{a}_i^T, b_i \right) \right\| + |c_{ji}| \|\lambda_i\|}{N_{\min}} \right] + \sum_{j=1}^{m} \sum_{i \in N_j} |c_{ji}| \left\| \left( \mathbf{a}_i^T, b_i \right) \right\|$$

$$\le \sum_{i=1}^{n} \left[ \frac{\sqrt{m} \left\| \varepsilon^{(i)} \right\| \|\lambda_i\|}{N_{\min}} + \left( 1 - \frac{1}{N_{\max}} \right) \sqrt{m} \left\| C_{:,i} \right\| \left\| \left( \mathbf{a}_i^T, b_i \right) \right\| \right] +$$

$$+ \sum_{i=1}^{n} \left[ \frac{\sqrt{m} \left\| \varepsilon^{(i)} \right\| \left\| \left( \mathbf{a}_i^T, b_i \right) \right\| + \sqrt{m} \left\| C_{:,i} \right\| \|\lambda_i\|}{N_{\min}} \right] + \sum_{j=1}^{m} \sum_{i \in N_j} |c_{ji}| \left\| \left( \mathbf{a}_i^T, b_i \right) \right\|$$

$$\le \sqrt{m} K \delta_{\max}^2 + \sqrt{m} \left( 1 - \frac{1}{N_{\max}} \right) \sum_{i=1}^{n} \left\| C_{:,i} \right\| \left\| \left( \mathbf{a}_i^T, b_i \right) \right\| +$$

$$\frac{\sqrt{m} \delta_{\max}}{N_{\min}} \sum_{i=1}^{n} \left( \left\| \left( \mathbf{a}_i^T, b_i \right) \right\| + \left\| C_{:,i} \right\| \right) + \sum_{j=1}^{m} \sum_{i \in N_j} |c_{ji}| \left\| \left( \mathbf{a}_i^T, b_i \right) \right\|$$

as desired. $\qquad\square$

