# OpenReview forum: "Neural Network Approximation based on Hausdorff distance of Tropical Zonotopes"
_ICLR.cc/2022/Conference — ICLR 2022 Poster_

### Official Review · Reviewer_mhFZ · 2021-11-02

**Correctness:** 3
**Technical Novelty And Significance:** 3
**Empirical Novelty And Significance:** 3
**Recommendation:** 6
**Confidence:** 2

**Main Review:**

Strengths:
1. This is a novel idea that has not been explored widely for structured network compression.
2. A proper introduction to tropical geometry and the related tool has been done for non-domain experts.
3. Illustrations are very helpful in understanding the method.

Weaknesses:
1. Comparison with non-tropical methods has not been performed.
2. Please report the wall clock time of finding a pruned network and inference time.
3. Without pseudo-code or code segment(s) for zonotope generators part, it is hard to reproduce the experiments. The paper will also be more accessible to non-domain experts if included. The experimental setup is very short and not enough to reproduce the results.

**Summary Of The Paper:**

In this paper, a novel method of structured pruning of neural networks using tropical geometry has been proposed. The core idea is to use the tools from tropical geometry to represent a neural network as tropical polynomials and then apply k-means clustering using some distance metrics that work on such representation.

The paper contains background information on tropical geometry and related tools. Experiments have been performed on linear layers of a binary classification network trained on MNIST (3/5 and 4/9) and LeNet5 and VGG trained on MNIST and Fashion-MNIST. The proposed method is compared with $L1$ structured pruning, Smyrnis & Maragos (2020), and ThiNet (Luo et al., 2017).

**Summary Of The Review:**

I am leaning towards acceptance of the paper as it provides a very interesting and new idea. As I am not a domain expert in tropical geometry, I would wait for other expert reviewers to judge the theoretical part. If authors can provide pseudo-code or code segment that makes this paper more accessible for me leading to better understanding, then I am willing to change my score.

---

> ### Author Response · Authors · 2021-11-19
> **Response to reviewer mhFZ**
>
> Thanks for pointing out our strengths. We are happy to hear that our work can be read by non-domain experts.
>
> Weaknesses.
>
> 1. We compare with Random and L1 structured as well as with ThiNet (all of them are non-tropical).
>
> 2. We do not intend to provide super-fast implementation as our experiments are proof-of-concept. We use sklearn K-means implementation and our code executes in descent time. For time complexity please see the general comment.
>
> 3. We provided the code as supplementary material.

---

### Official Review · Reviewer_FTWK · 2021-11-05

**Correctness:** 3
**Technical Novelty And Significance:** 3
**Empirical Novelty And Significance:** 3
**Recommendation:** 5
**Confidence:** 2

**Main Review:**

Questions:
- Why do the bounds decrease as the percentage of remaining weights decreases? This is pointed out but not explained. This suggests that the bounds of Prop. 4 and 5 aren't very useful. We should expect that if we can use more weights, we can at least avoid making the error worse, but this is not borne out by the computations in Table 2.

- Why was Alfarra et al. (2020) not compared to experimentally? The empirical comparison seems highly relevant. A discussion and comparison of the techniques should also be included since this work is quite related.

- What is deepNN? This is in the plot captions 4c and 4d but is never explained.

Feedback:
- The experimental results on CIFAR datasets are not very convincing. Compression of networks such as AlexNet and VGG is of limited interest as such architectures have been out-of-date for years. It would be much better to evaluate on modern architectures such as ResNets, Vision Transformers, etc.

- On MNIST and Fashion-MNIST, ThiNet generally does better except on Fashion-MNIST for LeNet.

- For all experiments, the authors should clearly state the current SOTA method for pruning and include those results as well. While beating SOTA is not necessarily a requirement, it at least needs to be provided for comparison.

- Also, the supplementary material was not submitted separately but instead included in the main PDF. While I actually prefer this approach for convenience, I believe they are supposed to be submitted separately. Also, it would be better to include the code for reproducibility.

Positive feedback:
- The background on tropical geometry is well-explained.

- Theorem 2 has a very clean statement. Q: Do related statements exist in the literature?

**Summary Of The Paper:**

This paper proposes to study neural network compression through the lens of zonotope reduction. They present approximation bounds for tropical polynomials based on Hausdorff distance and use this to motivate the development of two neural network compression methods. These methods are evaluated on the MNIST, Fashion-MNIST, CIFAR-10, and CIFAR-100 datasets.

**Summary Of The Review:**

This paper has some interesting results (such as Theorem 2) and the techniques are interesting and seem well-motivated (although this paper is not the first to study tropical geometric neural compression). However, the bounds from Propositions 4 and 5 seem to be not practically useful, and the experimental results have limitations. Thus, I hesitate to recommend acceptance at this time. It would be great if the authors can provide clear responses to my questions above.

---

> ### Author Response · Authors · 2021-11-19
> **Response to reviewer FTWK**
>
> Questions:
> 1. The decremental behavior of the bounds is because the bound is dominated by $K$ which experimentally occurs larger compare to the rest of the parameters.
> It could be an indication that the bound is loose, but this is still ok because it is the first step towards further research for finding stricter bounds.
>
> 2. We didn't compare to Alfarra et.al. because their method is unstructured (see general comment).
>
> 3. Thanks for pointing this out. We fixed this in the text.
> deepNN is a deep architecture consisting of 4 FCs layers and thus shows that our method is still efficient for deeper architectures.
>
> Feedback:
>
> 1. This is true. However, this is a first step towards providing a purely geometrical interpretation of neural network approximation. More complex architectures with different layers require further investigation with tropical geometry and will be our next step. In fact, ResNets are mainly convolutionally-based and have only one FC. Thus no practical compression can be achieved in our case.
>
> 2. On LeNet our method generally performs better, while on deepNN ThiNet is better. ThiNet was provided as an indication of our performance in comparison to the state-of-the-art. Our scope was not to claim that our method is better than ThiNet but to show that it can achieve similar results.
>
> 3. ThiNet is amongst the current SOTA for structured pruning (to our best knowledge - see general comment).
>
> 4. In the Author Guide of the conference it is stated that: *Supplementary (text) submission: We encourage authors to submit a single file (paper + supplementary text) this year. Please mark the supplementary material clearly.* Thus it shouldn't be a problem having them in the same pdf.
> We also added our code as extra supplementary material.
>
> Positive feedback
>
> 1. Thanks for pointing this out.
>
> 2. To our best knowledge, no similar statements exist in literature. It provides an interesting connection of the approximation error of two tropical polynomials with the Hausdorff distance of their Extended Newton polytopes. This connection also applies to neural network approximation. Note that we turned this theorem into a proposition and we use it to prove a more general theorem that works for neural networks. We would like to highlight that this new theorem is the main theoretical result of our work.

---

### Official Review · Reviewer_srEK · 2021-11-06

**Correctness:** 3
**Technical Novelty And Significance:** 3
**Empirical Novelty And Significance:** 3
**Recommendation:** 5
**Confidence:** 1

**Main Review:**

Pros: The proposed method seems novel and has theoretical guarantees.

Cons:
- The paper is a bit hard to process for general audience in machine learning, and the topic of the work does not seem to align well with ICLR. it would be nice if the authors could explain why this work would be good match to this conference.
- The emprical study is only performed on small scale datasets (a.k.a., MNIST and Fashion-MNIST). To compare with more modern compression techniques, experiments on more large scale datasets (e.g., CIFAR-10/100, COCO, Imagenet, etc) would better demonstrate the superiority of the proposed method.

**Summary Of The Paper:**

***This is a educated guess review as the paper is outside my domain expertise.

This paper proposes a compression method using a framework based on geometrical zonotope reduction. The authors further analyze the error bounds of the proposed methods and compare its performance with modern pruning techniques. I think the main contribution of the work would be novelty of the proposed method.

**Summary Of The Review:**

As this work is outside my domain of expertise, this is only an educated guess type of review, and I am willing to hear how other reviewers' opinions.

===Update after the authors' response===
The summary of the paper in the authors' response is very helpful for the reader to grasp the main message of this work. Still, I have decided to keep my score as a) the empirical contribution is a bit limited, and as mentioned by the authors, the experiments are mainly proof-of-concept; and b) though the theoretical contribution is novel, it still needs a bit more work to justify its significance.

---

> ### Author Response · Authors · 2021-11-19
> **Response to reviewer srEK**
>
> We hope to make our paper even easier to read for the general audience. For your better understanding, we provide the following summary of our paper structure.
>
> * Prior Work: Α layer of a network is represented by a set of geometrical structures called zonotopes.
>
> * Contribution: We deduce that compressing the zonotope leads to the function approximation of the network. This is theoretically deduced using tropical geometrical tools.
>
> * Contribution: For neural network compression, we select K-means algorithm
> and apply it to the zonotope generators (Zonotope K-means algorithm) or to the weights corresponding to the neurons (Neural Path K-means).
>
> * Contribution: We provide a geometrical approach for NN approximation and theoretical bounds on the error of the approximation of our algorithms and further study their performance empirically on compressing the FCs of known neural networks and datasets.
>
> Regarding your suggestions:
>
> 1. We provide a new theoretical method for reasoning about neural networks and both our theory and applications match with ICLR.
>
> 2. We agree. In fact, in our experiments, we use the CIFAR dataset. However, our experiments are proof-of-concept and it wouldn't make sense to go for larger datasets (COCO and ImageNet) unless we would like to propose a new state-of-art method.

---

### Official Review · Reviewer_ayne · 2021-11-07

**Correctness:** 3
**Technical Novelty And Significance:** 4
**Empirical Novelty And Significance:** 2
**Recommendation:** 5
**Confidence:** 3

**Main Review:**

### Strengths
1. The paper presents a mathematically solid analysis of neural networks (seen as tropical rational mappings), including a bound for the approximation error of the compressed network.
2. The presented algorithm seems to outperform a recent tropical baseline in terms of test accuracy of the pruned network.
3. Since the tropical viewpoint is a pretty new perspective on ReLU networks, the paper may inspire more interesting theoretical work in the area and may thus be of interest for theoretical researchers in the field.

### Weaknesses/Suggestions for Improvement
1. The CNNs used in the comparison in Figure 4 are quite old. It would be good to remove LeNet and have a more modern architecture, like a ResNet instead.
2. The paper neither mentions the computational and storage (memory) complexity nor the runtimes of the introduced algorithm. Having a comparison between the paper’s algorithm and the one from Smyrnis & Maragos would be beneficial.
3. Related to the previous point: All the networks used in the evaluation section are pretty small. Is it possible to run the algorithm on larger networks? Transformer networks contain only fully-connected layers, so it would be really interesting to apply the algorithm to them.
4. The paper states that the algorithm is only applicable to fully-connected layers, but it is tested on multiple CNN architectures in the experimental section. Are only nodes from the fully-connected layers pruned in the experiments? ThiNet is a filter pruning method, so I assume that at least some of the methods also prune the convolutional layers.
5. The paper should state in the abstract that the algorithms only apply to networks with only ReLU activations.
6. The last sentence of the abstract says: “We deduce that our methods (1) show an improvement over relevant tropical geometry techniques, (2) advance baseline pruning methods, and (3) have competitive performance against a modern pruning technique.” I agree with point (1): The proposed method seems to work better than the tropical one from Smyrnis & Maragos. Regarding (2): I cannot see how the method advances any baseline apart from the tropical one already mentioned in (1) - neither in terms of test accuracy nor in terms of speed. Regarding (3): The most “modern” pruning technique the authors compare to is ThiNet from 2017 which is clearly not state of the art by looking at Figure 3 of the pruning survey [1]. The statements in (2) or (3) should either be backed up by more concrete evidence or removed.
[1] What is the State of Neural Network Pruning? Blalock, Gonzalez Ortiz, Frankle, Guttag.

**Minor**

7. The captions to the tables should be self-explanatory. It is not clear from the captions that the figures in Table 1 and 3 are the test accuracies after pruning and that the figures in Table 2 are the left hand side of Proposition 5.


**Summary Of The Paper:**

The paper studies neural networks from the perspective of tropical geometry and applies its theoretical results to develop a novel algorithm for the compression of neural networks with ReLU activations. The presented algorithm outperforms a previous tropical network compression algorithm in terms of the accuracy of the compressed network.

**Summary Of The Review:**

The paper seems to be mathematically solid, but I see room for improvement in the experimental section. Since the experiments focus on small architectures, it is unclear to me whether the proposed algorithm can be used to compress larger networks. Since theoretical work has an important value on its own, the compression algorithm is not required to be of immediate practical interest. However, it should be made clear what the limitations of the presented method are. Judging from the presented experiments, the last sentence of the abstract may be overclaiming the experimental results. Nonetheless, the tropical geometry section seems intriguing and may be of interest to theorists.

---

> ### Author Response · Authors · 2021-11-19
> **Response to reviewer ayne**
>
> **Weaknesses/Suggestions**
>
> 1. Our experiments follow a proof of concept strategy. More complex architectures with different layers require further investigation with tropical geometry and will be our next step. In fact, ResNets are mainly convolutionally-based and have only one FC. Thus no practical compression can be achieved in our case.
>
> 2. In the general comment we note that our time complexity is basically derived by K-means. We claim that our complexity is smaller compared to Smyrnis & Maragos as their algorithm needs to use every sample of the training set, which in general can be very large, whereas our algorithm is independent of the dataset.
>
> 3. Larger networks usually have complex architectures and thus would require further tropical geometrical research to deduce compression methods for them. We are currently working towards compressing convolutional layers. Also, transformers would be an interesting extension and we will consider it as future work.
>
> 4. In all of our experiments we only compress FCs. ThiNet is also applied only to FCs.
>
> 5. We fixed this in the updated version.
>
> 6. We rephrased our last sentence in the abstract.
> Our experiments follow a proof-of-concept strategy in order to show that our theoretical claims indeed hold in practice. Neural Path K-means achieves competitive performance compared to non-tropical methods Random and L1 structured and ThiNet. Regarding ThiNet method please see the general comment. It may not be the state-of-art for all kinds of pruning methodologies, but it is amongst the state-of-art for the area of structured pruning.
>
> 7. Thanks, we fixed this.

---

### Author Response · Authors · 2021-11-19
**General Comments**

We would like to thank all the reviewers for their time spent understanding and appreciating our work. Their insightful comments and suggestions have helped us to improve the presentation of our paper. In particular, thanks to the reviewers’ comments, we realized that the scope of our paper, which is mainly a theoretical contribution and study, rather than a breaking-through pruning technique, is not clear to the reader. Thus, we made some changes and uploaded the revised version.

**Changes in our paper**
* We fixed all issues the reviewers pointed out. Namely, we rephrased the last sentence of the abstract, we mentioned deepNN in the text and we added in the abstract the information that we apply our methods on ReLU networks.
* We stated a **new theorem** that occurs from tropical polynomial approximation, and gives a bound for neural network approximation. This way we make more clear our contribution: with tropical geometry, we provide a theoretical bound on neural network approximation that depends on the Hausdorff distances of zonotopes.
* Wherever needed we changed the text to be compatible with the newly added theorem. Theorem 2 of tropical polynomials became a proposition that leads to the theorem of NN approximation. Then we also added this information to the abstract, the conclusion + the contributions section.
* We also changed the title of our paper to a more appropriate one.

**Scope of our work** We believe that with this change we make it more clear that we have a novel contribution to theoretically  support our geometric approach for neural network approximation. Our theoretical study is limited to networks with one hidden layer but our methods may be repeatedly applied for FCs of deeper networks. We thus make the first step towards a purely geometrical interpretation of neural network approximation. Also, our experiments follow a proof-of-concept strategy in order to show that our theoretical claims indeed hold in practice. Thus we do not aim to provide a better method than the State of Art.


The following general comments apply to the reviewers’ inquiries.

**Structured vs Unstructured + State of Art** We need to make clear that our method lies in the area of structured pruning i.e. remove whole neurons from the network which differs from unstructured pruning that sparsifies the weight matrices of the layers. It is thus not beneficial to compare structured methods with unstructured due to their distinguished nature. It would not be informative from a theoretical viewpoint.

For example, Alfarra et.al. provide a tropical compression method that is basically an optimization problem for sparsifying the layers matrices and thus unstructured. Further, their compression attempts to preserve the decision boundary, rather than perform a function approximation of the network as we do. The former can only be applied to classification problems, whereas our method is also applicable for regression problems.

It is worth also stating that even unstructured pruning may achieve higher compression performances, it is in fact impractical for existing hardware systems that are incapable of attaining considerable speedup for arbitrary sparsity forms. Thus working towards the research direction of structured pruning is valuable.

ThiNet method is a structured pruning method that is amongst the state-of-the-art in this area. We thus provide a comparison to it to have an indication of our performance compared to the SOTA. We thus do not aim to compare our methods exhaustively with existing SOTAs methods which have been matured through years of development and research, whilst we try to introduce a novel geometrical approach.


**Code + Memory/Runtime Complexity** For our experiments to be reproducible we provide our code as supplementary material. We will also make our code publicly available through Github.
The runtime complexity of our algorithm is dominated by the runtime of K-means which is essentially is $\mathcal{O}(InKD)$ where $I$ is the number of iterations needed for K-means, $n$ is the number of hidden nodes, $D$ is the dimension of the vectors and $K$ the number of final nodes (clusters). The memory complexity also coincides with K-means $\mathcal{O}((n+K)D)$.

For Zonotope K-means $D = d + 1$ which is essentially the input layer dimension plus bias and for Neural Path K-means $D = d + 1 + m$, where m is the output layer dimension.

---

### Decision · Program_Chairs · 2022-01-20

**Decision:**

Accept (Poster)

**Comment:**

The submission introduces an algorithm for structured pruning of fully connected ReLU layers using ideas from tropical geometry. The paper begins with a very accessible overview of key concepts from tropical geometry, and shows how ReLU networks can be thought of as  tropical polynomials. It gives an efficient K-means-based algorithm for pruning units in a way that approximately minimizes the Hausdorff distance between certain polytopes. Experiments show that the method outperforms other methods based on tropical geometry and is competitive with SOTA methods from a few years ago.

I think the reviewers, authors and I all agree on the following points: tropical geometry is a mathematical topic not commonly used in our field and for which it is difficult to find expert reviewers (notice that most of the citations aren't from ML venues). The paper is well-written, and the authors have taken pains to present the required concepts in an accessible way. Nobody has raised any concerns about correctness. While this isn't the first pruning method that uses tropical geometry, the algorithm is novel and interesting. It's expensive, but not unreasonably so. The experiments are a proof-of-concept: they use small networks by today's standards, and the baselines aren't the current SOTA.

The average scores are slightly below the usual cutoff. The reviewers are concerned about whether this method is useful, given that is based on different principles from current methods and can't quite compete with current SOTA. But my own sense is that this is a paper that we'd like to have at ICLR. It gives a clear, accessible introduction to tropical geometry and demonstrates its usefulness for practical deep learning. It demonstrates competitiveness with fairly strong baselines, which is all we should expect from methods that haven't benefited from years of hill-climbing on the same handful of ideas. I recommend acceptance.